# Effects of Decentral Heat Pump Operation on Electricity Storage Requirements in Germany

**Simon Hilpert** 

Department of Energy and Environmental Management, Europa Universität Flensburg, 24941 Flensburg, Germany; simon.hilpert@uni-flensburg.de; Tel.: +49-461-805-3067

**Abstract:** Several studies show that heat pumps need to play a major role for space heating and hot water supply in highly decarbonised energy systems. The degree of elasticity of this additional electricity demand can have a significant impact on the electricity system. This paper investigates the effect of decentral heat pump flexibilisation through thermal energy storage units on electricity storage investment. The analysis is carried using an open source model for the German electricity system based on the Open Energy Modelling Framework (oemof). Results highlight the importance of flexible heat pump operation in 100% renewable energy systems and relate well to findings of other existing studies. Flexibilisation of heat pumps in the German energy system can reduce the need for electricity storage units significantly. While no impact was found for systems with a share below 80% renewable energy, investment in short term storage units is reduced by up to 42–62% in systems with shares of more than 80% renewable energy. In contrast, the impact on long term electricity storage investment was comparatively low in all modelled scenarios. Conducted sensitivity analyses show that both findings are rather insensitive with regard to the available biomass for electricity supply as well as to changes in the heat demand covered by heat pumps. Economically flexible heat pump operation has only a minor effect on system costs. However, the indirect replacement of battery with thermal energy storage units is environmentally beneficial due a lower resource consumption of minerals.

**Keywords:** energy system modelling; 100% renewable energy systems; open science; sector coupling; heat pump; flexibility options; thermal energy storage; electricity storage

## 1. Introduction

The goal of the 2015 Paris agreement [1] is to keep global warming well below two degrees compared to pre-industrial levels. In 2018, the special report of the Intergovernmental Panel on Climate Change (IPCC) [2] reaffirmed the importance of this goal by analysing pathways for a warming of 1.5°. Due to the remaining carbon budget, a drastic decarbonisation up 100% of all sectors until 2050 with even negative emissions after the year 2050 will be required to reach the 1.5° goal. In the electricity sector emissions are mainly reduced through a shift from fossil fuel based to renewable energy based supply. Within the heating sector, this solution is rather challenging as renewable resources are limited. Therefore, reducing energy consumption in the heating sector by insulating measures is on the top of the agenda. Nevertheless, a residual heat demand will have to be covered by renewable energies. District heating (DH) systems allow for better integration of renewable technologies compared to individual heating systems. However, the DH potential is also limited as systems require certain spatial heat demand densities for economic operation.

For individual heating solar thermal, biomass or electricity are left as the major options in Germany. Solar thermal energy has to cope with opposed seasonality of demand and supply. Hence, only small shares of solar thermal energy may be integrated in the heating sector without seasonal storage units.

Therefore, the economic potential of solar thermal supply in Germany is limited to around 60 TWh$_{th}$ annually [3]. Energy production from sustainable biomass conflicts with nature conservation and food production. In addition, the heating sector and the transport sector (aviation and shipping) will compete with one another in carbon neutral societies due to the high value of transportable and storable energy (see discussions in References [4–7]). Finally, heat pumps are an energy efficient option to supply heat and reduce $CO_2$ emissions [8,9]. Due to the above mentioned reasons, the authors argue that "[...] heat pumps are deemed the most suitable individual heating solution in a 100% renewable energy system for the EU" [10], p. 1644. In various studies for highly decarbonised energy systems heat pumps and solar thermal collectors are the dominant energy sources [11–13]. Especially for individual heating systems this technology is often mentioned as the major option in Germany [14,15]. A broad roll-out of decentralised heat pumps moves decarbonisation challenges from the heating sector to the electricity system. A central question regarding the added electricity demand induced by heat pumps is their elasticity to match with intermittent renewable energy supply. Elasticity of heat demand may be increased by a thermal energy storage (TES) with a positive impact on the electricity system. The aim of this paper is to analyse the effect of decentral heat pump flexibilisation on electricity storage investment in renewable energy systems.

## 2. State of Art and Research Question

Bloess et al. [16] review power to heat technologies for renewable integration. The authors conclude that sector coupling comes with multiple benefits such as a reduced peak load, lower electricity storage needs, less renewable curtailment and more efficient power plant dispatch. For the mid-term perspective the Danish energy system in 2030 is optimised with the open source *Balmorel* model by Hedegaard and Münster [17]. Results suggest great importance of residential decentralised heat pumps for the integration of wind energy. However, only a minor effect of flexibilisation through TES is observed. At the European level, Brown et al. [11] analyse synergies of sector coupling in highly decarbonised energy systems with an open source investment model based on the Python package *PyPSA*. The heating scenario of this study shows a positive effect of long and short term thermal energy storage (TES) for integrating solar thermal heat as well as thermal energy from power to heat units. Heat pumps play a significant role for decentralised heat supply, that is, in areas with low density of heat demand where district heating is not a reasonable option. For the electricity-heat coupling long term storage units contribute significantly to integrating Wind and PV. However, no detailed analyses of HP flexibilisation in systems under different renewable energy penetration and different heat demands are provided in this study. Also, the power-to-energy ratios are fixed in this model. Hence, no statement on required optimal storage energy capacity can be given. An analysis for Germany in the European context is presented by Bernath et al. [18] to investigate the role of heat pumps for renewable energy integration using the optimisation tool *Enertile*. The closed source model includes district as well as decentral heating systems with heat pumps. Ruhnau et al. [19] analysed the effect of heat pumps on the economic value of wind with the open source market model *EMMA* for Germany. The modelled scenarios also include an analyses of interdependencies between different flexibility options that indicate lower electricity storage investment due to the existence of thermal storage capacities in scenarios with 30% wind energy supply. Fehrenbach et al. [20] optimised the residential German heating sector under varying levels of renewable energy expansion using a *TIMES* model. Unfortunately, model source code and data for this study are not publicly available. In addition, the overall optimisation approach does not allow to compare effects of inflexible and flexible operation. The impact of increased power-to-heat on the heat sector transformation in Germany is also analysed by Bloess [21] with a multi-period expansion model. This study models different levels of heat demand with and without power-to-heat and determines a major impact of power-to-heat on the electricity sector. The author concludes that thermal storage plays a greater role than short term electricity storage, although further verification is required. Many studies have investigated electrical storage requirements on the European and German level [22–25]. These studies solely focus on the

electricity sector and do not analyse the interdependencies between flexibility options in the heat and electricity market.

The literature review shows that a number of relevant studies for sector coupling and heat pumps are available. Nonetheless, no open source modelling approach exists to analyse the effect of heat pump flexibilisation in settings with different renewable shares. Specifically the interdependencies with other flexibility options are not assessed in detail by a ceteris paribus approach. This paper investigates the interactions of flexibility options in an electricity-heat sector coupled system. In particular the impact of heat pump flexibilisation through TES in the decentral heating sector is analysed, regarding its influence on electricity storage investment and operation. The analysis is conducted based on an open source model for the German energy system including the neighbouring countries.

Subsequently, Section 3 provides a mathematical description of the model followed by an overview of modelled scenarios with their relevant input data in Section 4. Based on these two sections results are presented in Section 5. The last section provides a short summary followed by a critical appraisal of the study.

## 3. Method

Lund et al. [26] describe differences between two methodological positions: simulation vs. optimisation. In this paper a hybrid approach is chosen to analyse the effects of heat pump flexibilisation. While installed generation capacities and the transmission grid capacities are defined exogenously, storage and heat pump capacities are determined endogenously by optimisation. With this approach, effects of heat pump flexibilisation on electricity storage units can be assessed without interference of other system variables. The analysis is carried out with a linear programming optimisation model based on the Python package *oemof-tabular* which is part of oemof cosmos [27]. The source code of the package is available on GitHub [28] under the BSD 3-Clause license.

Figure 1 illustrates the graph based model of a power and heat coupled energy system with this software.

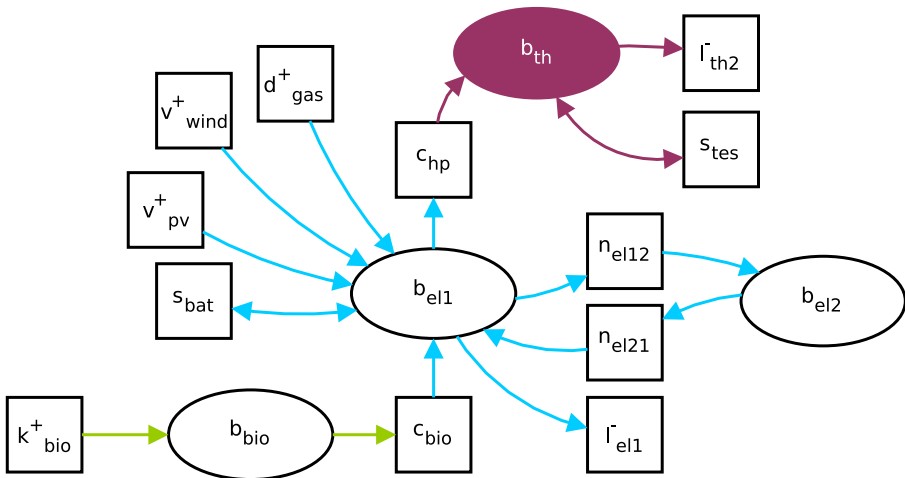

**Figure 1.** Illustration of a sector coupled energy system modelled based on oemof tabular. The energy system is modelled as a bi-partite graph with components (squares) and buses (ellipses). Electricity flows are coloured blue, biomass green and heat red.

*Mathematical Description*

The underlying mathematical model of this graph structure is implemented in the *oemof-solph* package. In the following, mathematical description of all endogenous variables are denoted by $x$, while all exogenous variables are denoted by $c$.

The model is a combined dispatch and investment model with exogenously defined parameters for the electricity system and investment for electricity storage units and the decentral heating system (HP and TES). For the investment part of the model all dispatch constraints below apply as well. However, the upper bounds of the maximum capacity of HP, TES and electricity storage units (except PHS) are subject to optimisation. The objective function of the model minimises total operating and investment costs, as shown in Equation (1).

$$
\text{min:} \sum_g \sum_t \overbrace{c_g^{mc} \cdot x_g^{flow}(t)}^{\text{operating cost}} +
$$

$$
\sum_h \overbrace{c_h^{capacity\_cost} \cdot x_h^{capacity}}^{\text{investment cost HP}} +
$$

$$
\sum_s c_s^{capacity\_cost} \cdot x_s^{capacity} + \overbrace{c_s^{energy\_cost} \cdot x_s^{storage\_capacity}}^{\text{investment cost storage}}. \tag{1}
$$

The marginal costs $c_g^{mc}$ of a generator $g$ are calculated based on carrier $c_g^{cc}$ costs, variable operation and maintenance $c_g^{vom}$ cost and $CO_2$ costs $c_g^{co2}$ that are determined based on the carrier specific emission factor of the generator $e_g^{carrier}$ (Equation (2)).

$$
c_g^{mc} = \frac{c_g^{cc}}{\eta_g} + c^{co2} \cdot e_g^{carrier} + c_g^{vom}. \tag{2}
$$

The investment costs are defined as the annualised capacity costs including fixed operation and maintenance (fom) costs. For storage units, these costs are composed of an energy and a power component. For the TES, the power-energy-ratio is not fixed to determine the optimal TES sizing.

**Energy balances** and **commodity balances** are modelled with the set of Buses $B$. For buses all inputs $x_{i(b),b}^{flow}$ to a bus $b$ must equal all its outputs $x_{b,o(b)}^{flow}$ (Equation (3)).

$$
\sum_i x_{i(b),b}^{flow}(t) - \sum_o x_{b,o(b)}^{flow}(t) = 0 \qquad \forall t \in T, \forall b \in B. \tag{3}
$$

Equation (4) shows the constraint for inelastic loads. For the set of all **loads** denoted with $l \in L$ the load $x_l$ at time step $t$ equals the exogenously defined profile value $c_l^{profile}$ multiplied by the total annual demand $c_l^{demand}$

$$
x_l^{flow}(t) = c_l^{profile}(t) \cdot c_l^{demand} \qquad \forall t \in T, \forall l \in L. \tag{4}
$$

**Dispatchable units** ($d \in D$) such as fossil fuel based power plants are limited by the defined capacity (Equation (5)). Marginal costs of the generators are calculated based on Equation (2) and added to the objective function.

$$
x_d^{flow}(t) \leq c_d^{capacity} \qquad \forall t \in T, \forall d \in D. \tag{5}
$$

**Volatile renewable supply** is modelled as must-run production. For all volatile components denoted with $v \in V$ the flow is fixed as described in Equation (6). The set of all volatile components includes all volatile sources.

$$
x_v^{flow}(t) = c_v^{profile}(t) \cdot c_v^{capacity} \qquad \forall t \in T, \forall v \in V. \tag{6}
$$

**Biomass units** and **Heat pumps** are modelled with a conversion process of one input and one output and a conversion factor shown in Equation (7).

$$x_{c,to}^{flow}(t) = c_c^{efficiency}(t) \cdot x_{c,from}^{flow}(t) \qquad \forall c \in C, \forall t \in T. \tag{7}$$

In the case of biomass plants the outflow is exogenously bounded by its nominal power rating as it is modelled for other dispatchable units. For the set of heat pumps $h \in H$ the flow is bounded by an optimisation variable $x_{h,to}^{capacity}$ shown in Equation (8).

$$x_{h,to}^{flow}(t) \le x_{h,to}^{capacity} \qquad \forall h \in H, \forall t \in T. \tag{8}$$

In combination with the commodity components (Equation (9)), the biomass supply can be limited by the available biomass potential by setting an upper limit on the aggregated flow of the component. The variable $x^{flow}k$ represents inflows for a biomass commodity bus from which the conversion process is fed.

$$\sum_t x_k^{flow}(t) \le c_k^{amount} \qquad \forall k \in K. \tag{9}$$

For **storage** units ($s \in S$), the mathematical representation includes the flow into and out of the storage as well as a filling level. The inter-temporal energy balance of the storage is given in (10). The loss rate for the storage can be obtained by a time constant $loss\_rate = 1 - \exp^{-\frac{1}{24 \cdot d}}$, where $d$ denotes the time constant in days.

$$x_s^{level}(t) = x_s^{level}(t) \cdot (1 - c_s^{loss\_rate}) - \frac{x_{s,out}^{flow}}{c_s^{eta\_out}} + c_s^{eta\_in} \cdot x_{s,in}^{flow}(t) \qquad \forall t \in T, \forall s \in S. \tag{10}$$

For the storage technologies with investment, the out- and inflow $x_{s,*}^{flow}$ as well as the energy $x_s^{level}$ is bounded by an optimisation variable (Equations (11) and (12)).

$$x_{s,in}^{flow}(t) \le x_s^{capacity} \qquad \forall t \in T, \forall s \in S \tag{11}$$

$$x_s^{level}(t) \le x_s^{storage\_capacity} \qquad \forall t \in T, \forall s \in S. \tag{12}$$

**Hydro reservoirs** are modelled as storage units with a constant inflow and possible spillage described in Equation (13).

$$x_r^{level}(t) = x_r^{level}(t-1) \cdot (1 - c_r^{loss\_rate}(t)) + x_r^{profile}(t) - \frac{x_{r,out}^{flow}(t)}{c^{efficiency}(t)} \qquad \forall t \in T, \forall r \in R. \tag{13}$$

The inflow is bounded by the exogenous inflow profile (Equation (14)). Thus, if the inflow exceeds the maximum capacity of the storage, spillage is possible by setting $x_r^{profile}(t)$ to lower values. The spillage of the reservoir is therefore defined by $c_r^{profile}(t) - x_r^{profile}(t)$.

$$0 \le x_r^{profile}(t) \le c_r^{profile}(t) \qquad \forall t \in T, \forall r \in R. \tag{14}$$

**Transmission** between the countries is modelled with a transshipment approach, as shown in Equation (15).

$$x_{from,n}^{flow}(t) = (1 - c_n^{loss}) \cdot x_{n,to}^{flow}(t) \qquad \forall n \in N, \forall t \in T. \tag{15}$$

**CO$_2$-emission** limit $\overline{L_{CO_2}}$ is set for all flows $x_e^{flow}$ indexed by $e \in E$ with by Equation (16).

$$\sum_t \sum_e x_e^{flow}(t) \cdot c_e^{emission\_factor} \leq \overline{L_{CO_2}}. \tag{16}$$

## 4. Scenarios

Systems with different renewable energy shares of 60% in 2030 to 100% in 2050 have been implemented within the described model to analyse the impact of heat pump flexibilisation with TES in Germany. The scenarios are based on the TYNDP2018 for the years 2030 and 2040 and on the e-Highway scenario 100% RES [29]. The following section provides an overview about important scenario assumptions. The source code of the model including script for generating input data can is publicly available on Github [30].

The model covers Germany with its electrical neighbours applying a spatial resolution of one node per country. A temporal resolution of one hour is chosen with a total time horizon of one year. The grid capacities are taken from the e-Highway and TYNDP2018 databases (s. Appendix B.3). For the neighbouring countries of Germany published data on installed capacities from the TYNDP2018 [31] as well as the e-Highway [29] project have been used. This data has also been used for commodity cost and operational expenditures. Adaptions have been made for Germany with regard to the installed capacities as well as the electricity and heat demand assumptions (Sections 4.1 and 4.2 below). With these adaptations the 2050 scenario represents the Green Late (GL) scenario of the RESCUE study for Germany [14].

### 4.1. Installed Capacities

Figure 2 shows the installed capacities in Germany and the respective renewable energy share for each scenario.

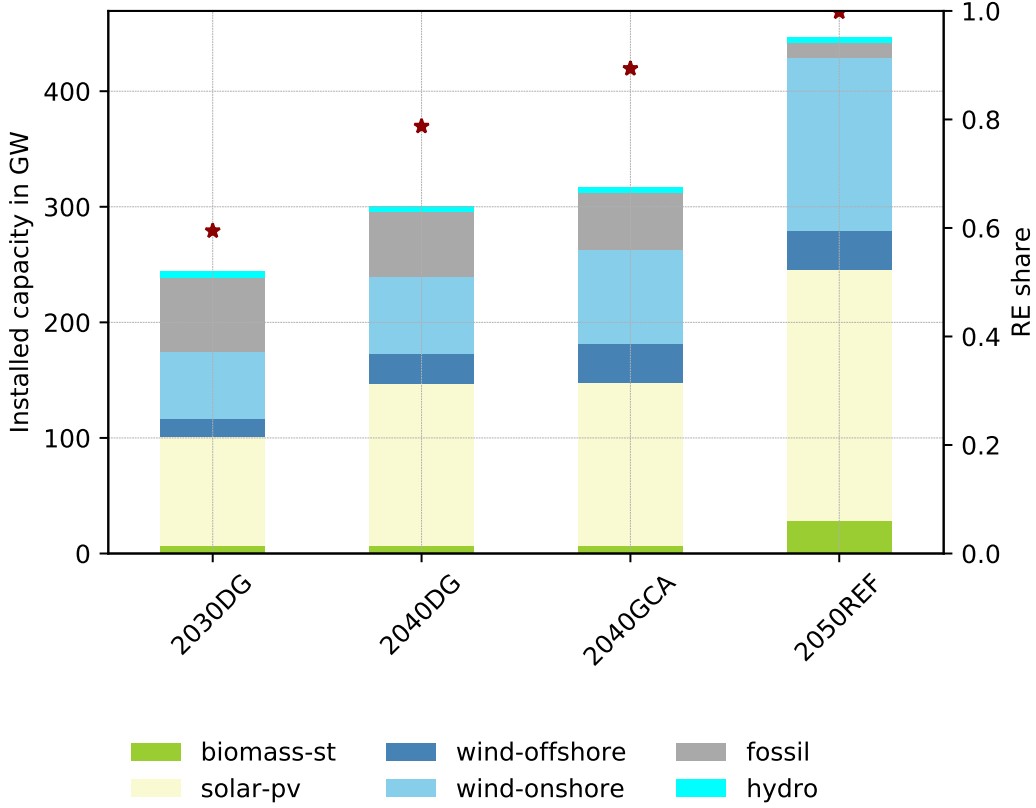

**Figure 2.** Installed generation capacities in Germany (DE) for all scenarios.

The *2030DG* scenario has the lowest installed capacities of renewable energies resulting in the lowest renewable energy share of around 60%. In 2040 the most progressive scenario is the *2040GCA* with a renewable share of approx. 88%. Compared to the *2040DG* this scenario has more wind onshore and offshore capacity installed. Also other European countries have a higher share of renewable energy in their energy mix. The *2050REF* scenario is based on the e-Highway 2050 100% RES (Europe) and the RESCUE 2050 GL scenario (Germany) and results in a scenario with 100% RE supply.

*4.2. Demand*

Assumptions regarding the electricity demand are a driving factor for the energy system. Values are associated with a high degree of uncertainty as the development of the future electricity demand strongly depends on demographic and economic development as well as implemented policy measures. The German goals regarding efficiency aim to reduce the conventional electricity demand (i.e., excluding electric vehicle, power-to-heat) by 10% until 2020 and 25% by 2050 compared to 2008 levels (403.8 TWh). Within the *Basis Szenario* of the German *BMWI Langfristszenarien* the conventional net electricity demand accounts for 417.2 TWh in 2050. The total gross electricity demand accounts for 612.4 TWh [32], p. 221. In the e-Highway 100% RES scenario the total gross demand is 665 TWh. Other studies suggest considerable higher electricity demand levels for 100% systems. For example, Reference [33], p. 9 model systems with a demand higher than 1000 $TWh_{el}$ and over 200 $TWh_{el}$ of excess energy in some scenarios. This shows the great range of possible future electricity demand levels. For the scenarios of this study the inelastic electrical demand (demand excluding heat pump consumption) has been based on scenarios for electrification of other transport and heat sectors according to the RESCUE [14] study to match with the installed generation mix. The demand calculations are shown in Table 1. For non-German countries, data from TYNDP2018 and the e-Highway project has been used. Normalised time series for electricity load profiles have been generated with the OPSD project data [34].

**Table 1.** Electricity demand values are based on the German efficiency goals. For the GS scenario it is assumed that a reduction of 25% and for GL 15% reduction is achieved.

|  | 2030 | 2040 | 2050 |
|---|---|---|---|
| Reduction (2008) | 0.10 | 0.125 | 0.15 |
| Electricity demand in TWh | 485 | 471 | 458 |
| Transportation (EVH) in TWh [14] | 30 | 80 | 115 |
| **Demand in TWh$_{el}$** | 515 | 551 | 573 |
| Distribution Loss [32], p. 221 | 0.11 | 0.09 | 0.07 |
| **Demand incl. losses in TWh$_{el}$** | 571 | 601 | 613 |
| **Heat covered by HP in TWh$_{th}$ [14]** | 57 | 195 | 284 |

Total heat demand per year is based on the RESCUE scenarios, which describe $CO_2$ neutral energy systems in 2050. In the selected *GreenLate (GL)* scenario heat demand covered by heat pumps accounts for 57 $TWh_{th}$ (2030), 195 $TWh_{th}$ (2040) and 284 $TWh_{th}$ (2050) [14]. Values for decentral heat production from heat pumps of the *RESCUE GL* scenarios are in the range of scenarios described in Hansen et al. [33]. Compared to [18] the additional electricity demand induced by decentral heat pumps is considerably higher in the RESCUE scenarios. To examine impacts of different heat demand levels a sensitivity analysis for the heat load is conducted. For the normalised heat profiles of hot water and space water heating another OPSD data set [35] has been used. The same data set has also been used to model the variable COP of the HP.

*4.3. Investment Data*

Pumped hydro storage (PHS) capacities have been set endogenously as their potential is strongly limited. For additional storage investment two different types of storage units are modelled.

One representing a short term storage option (lithium battery) and another representing a long-term storage (hydrogen storage) option. The parameters for the storage and heat pump investment are shown in Table 2.

**Table 2.** Data for the decentral heating system based on [11], data for the electricity storage based on [23,36,37]. Storage efficiency is shown as round trip efficiency.

| | Investment Cost | | FOM | Lifetime | WACC | Efficiency | Storage Capacity |
|---|---|---|---|---|---|---|---|
| | Euro/kW | Euro/kWh | Euro/kW(h)a | Years | | | h |
| **HP** | 1400 | - | 49 (Value in Euro/kWa) | 20 | 0.05 | variable | - |
| **TES** | 0 | 38.4 | 0.39 | 20 | 0.05 | 0.81 | endogenous |
| **Lithium 2050** | 35 | 187 | 10 | 20 | 0.05 | 0.92 | 6.5 |
| **Hydrogen 2050** | 1000 | 0.2 | 10 | 22.5 | 0.05 | 0.46 | 168 |

### 4.4. Renewable Generation

The solar PV and onshore wind profiles are based on the *renewables ninja* project [38,39]. Run of river profiles have been calculated with results of the Restore2050 project [40]. The total inflow provided in the data set has been split in proportion to the run of river and reservoir capacities in the scenarios. The weather year 2011 has been selected for all scenarios [11]. The full load hours of the volatile energy supply for different renewable technologies and each country are given in the appendix in Table A4.

The maximum biomass potential per country is derived from the *hotmaps* project [41] and is equal among all scenarios. The potential does not cover waste but only agriculture and forestry residues. For Germany the available potential has been adapted to values of the RESCUE study. With an electrical efficiency of 48.7 % for biomass to electricity conversion the potential in Germany is around 22 TWh$_{el}$ (s. Appendix B).

## 5. Results

The following section presents the results of the modelled scenarios. First the optimal investment in storage units is presented. Results of the sensitivity analyses are described at the end of this section.

### 5.1. Heating System Investment

Figure 3 shows the results for the investment in the heating system. The interaction between electricity system and heat pump operation can already be identified in this figure. The main driver for the investment in TES is the heat demand. For the 2030 scenario with low heat demand covered by heat pumps and a lower share of renewable energy no investment in TES is chosen.

In all scenarios with renewable energy shares above 80% the installed energy capacity ranges from around 108 TWh to around 150 TWh. No additional investment into heat pumps above their lower bound of the heat peak-load demand is observed. The optimal sizing of TES for the covered heat demand of 284 TWh$_{th}$ in 2050 is around 150 GWh. With the area for space heating of the GL scenario, this would amount for 0.5 L/m$^2$ water tank volume of the heated space area (6.37 Mrd m$^2$ in the GL scenario). The TES investment of 108 to 152 TWh$_{th}$) for all scenarios above 80% RE share is in the range of results determined by [20] (52–252 TWh$_{th}$).

For all scenarios the energy of the storage is in a range of 1.2–1.4 times the respective thermal peak load. Interestingly, this value is in line with current practices of storage sizing in (district) heating systems [42,43]. The power-to-heat ratio of the TES is significantly lower than the assumed values of 72 h of maximum installed capacity in MW$_{th}$ by Reference [11]. The reason for this difference can be found in the low investment cost per MW$_{th}$. With low cost per MW$_{th}$, the optimal values shift to higher capacities even though only a marginal return on investment exists.

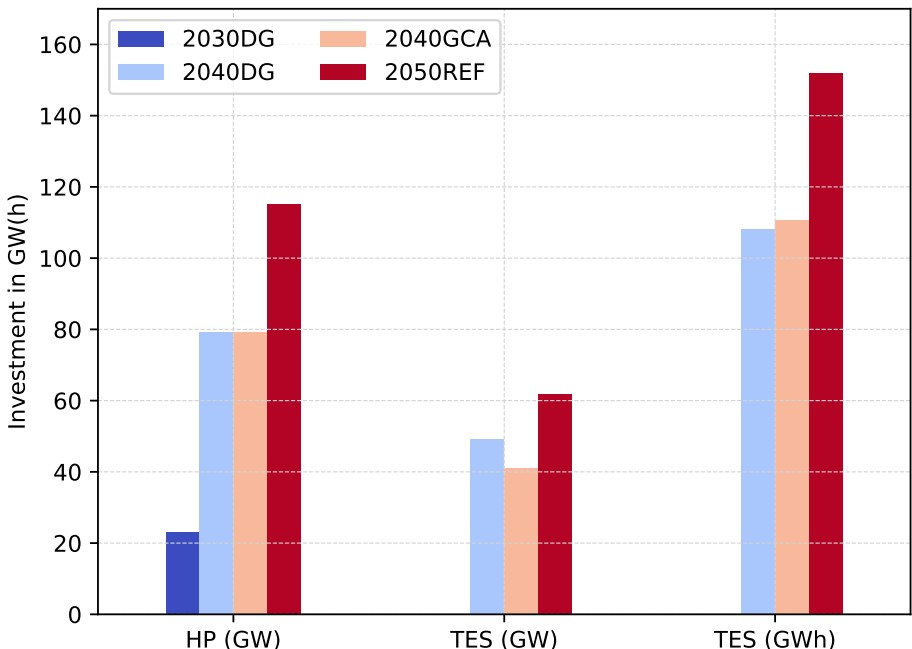

**Figure 3.** Investment in decentral heating system components for different scenarios. Units are given in GW and GWh.

### 5.2. Electricity Storage Investment

Figure 4 shows the investment in short-term (lithium) and long-term (hydrogen) storage units for the different scenarios. The investment increases with higher shares of renewable energy. For the *2030DG* system no additional storage besides PHS is required in Germany. Long term storage investment can only be observed in the 2050 scenario.

The results show that the effect of heat pump flexibilisation is significantly higher for short term storage units. Obviously, for the scenario where no TES is installed, no change in electricity storage capacity can be observed. Flexibilisation of HP by TES can reduce short-term storage investment by 3.3 GW (42%) up to 5 GW (61%) . For the long term storage, investment is only decreased by 0.37 GW (6.8%) in the 2050 scenario.

The reduction in short term electricity storage investment induced by heat pump flexibilisation matches with results from [19], where PHS investment can be reduced by around 5 GW in a scenario with 30% wind share. Results for the *2050REF* scenario (12.8 PHS and almost 11 GW lithium battery) are also comparable with 21 GW of short term storage requirement in 100% systems in Germany of [23]. However, a highly flexible heating sector can reduce additional investment by around 4.9 GW (44.5%). The authors of the "storage roadmap study" [25] highlight the great range of storage investment and their dependence on driving system variables like biomass potential and demand side management (DSM). In their study, DSM can reduce storage demand from 19.2 (no DSM) to 5.5 GW (max. DSM) for a system with around 88% RE-share in Germany, i.e., by around 71% in Germany [25], p. 88. The short-term storage requirements within these scenarios are also similar to the *2040GCA* scenario with 88% RE-share. For a 100% RE-system in Germany Müller et al. [37] identify a total storage investment of 13.7 GW (excluding PHS) with a majority of the investment found in hydrogen storage units. In contrast to this paper, their model includes an intra-country grid constraint. Grid bottlenecks can cause higher long-term storage investment to integrate (offshore) wind supply, which is indicated by investment in northern Germany. Due to the copper-plate approach in the presented model in this study, such bottlenecks can not be reflected.

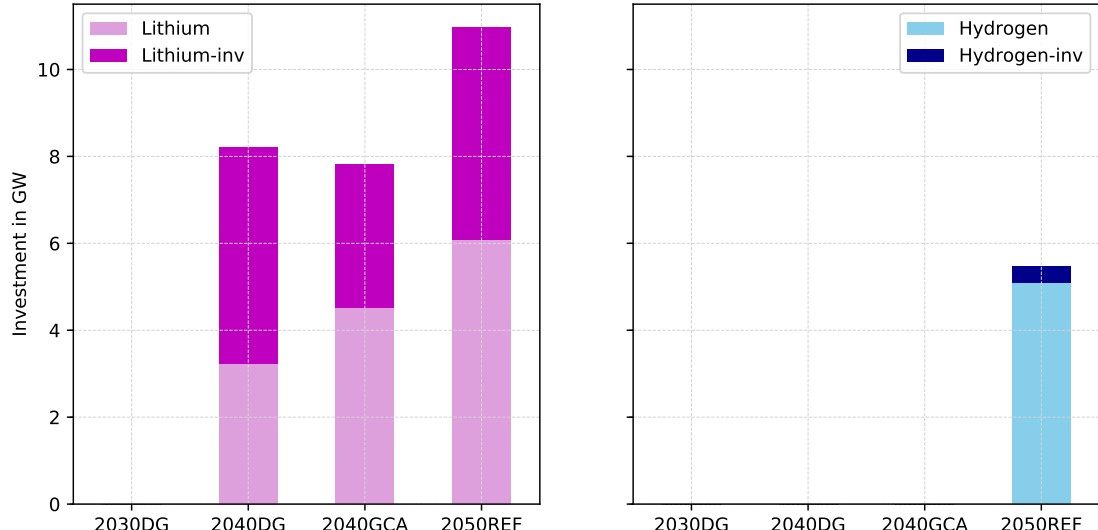

**Figure 4.** Investment in electricity storage units in GW with lithium on the left and hydrogen on the right side. The power has been chosen for better representation. Energy can be calculated based on the assumption of the maximum storage capacity in hours (6.5 h for lithium and 168 h for hydrogen). The absolute height of the bars represent investment without TES, that is, heat pump flexibilisation. Light dyed part represents electricity investment with HP flexibilisation. Therefore, dark coloured parts of the bars represent investment required due to inflexible heat pump operation.

### 5.3. System Costs

Table 3 lists the objective values for four scenarios. Economically, only small changes can be observed due to the flexibilisation. For the 2050 scenario the total costs are reduced by 0.52% in the case of elastic heat pump electricity demand compared to an inelastic demand. The lowest reduction with 0.27% takes place in the *2040DG* scenario.

**Table 3.** Objective value for scenarios with (Flex) and without (No-Flex) flexibilisation. Deviation may occur due to rounding of values inside the table.

|  | No-Flex (bn Euro) | Flex (bn Euro) | Change (%) |
|---|---|---|---|
| 2050REF | 22.32 | 22.20 | 0.52 |
| 2040DG | 41.49 | 41.38 | 0.27 |
| 2040GCA | 26.80 | 26.71 | 0.34 |
| 2030DG | 45.38 | 45.38 | 0.00 |

### 5.4. Storage Dynamics

Figure 5 presents a closer look on lithium battery (b) and the TES storage (b) cycles for the 2050 scenario. For cycle counting the Python package CyDeTs [44] has been used. The plot shows that the majority of cycle length are below the value of 72 h with a clear peak around 24 h and a smaller peak at around 10 h. The pattern of the electricity and the TES storage are similar. Note that this is not forced by the same underlying mathematical model approach as the ratio between storable energy and capacity of the TES, which has not been fixed inside the optimisation.

A majority of time, the storage units operate at full cycles (DoC of 1). Two different cycle length occur due to different operations in winter and summer time. In winter, shorter cycles are used to integrate PV peaks and shift energy a few hours towards the evening. In contrast, shifts in summer can be used to meet demand of longer periods of time during the night. The analysis of TES storage cycles shows that a fixed ratio of 72 h proofs as a reasonable assumption for systems with high shares of renewable energies. If a complexity reduction of models is required, results can also be used as

an indication for temporal aggregation measures to reduce computational run times of large models. Here, aggregating data on a daily basis will reflect the basic pattern of storage dispatch as most cycles of the TES are included.

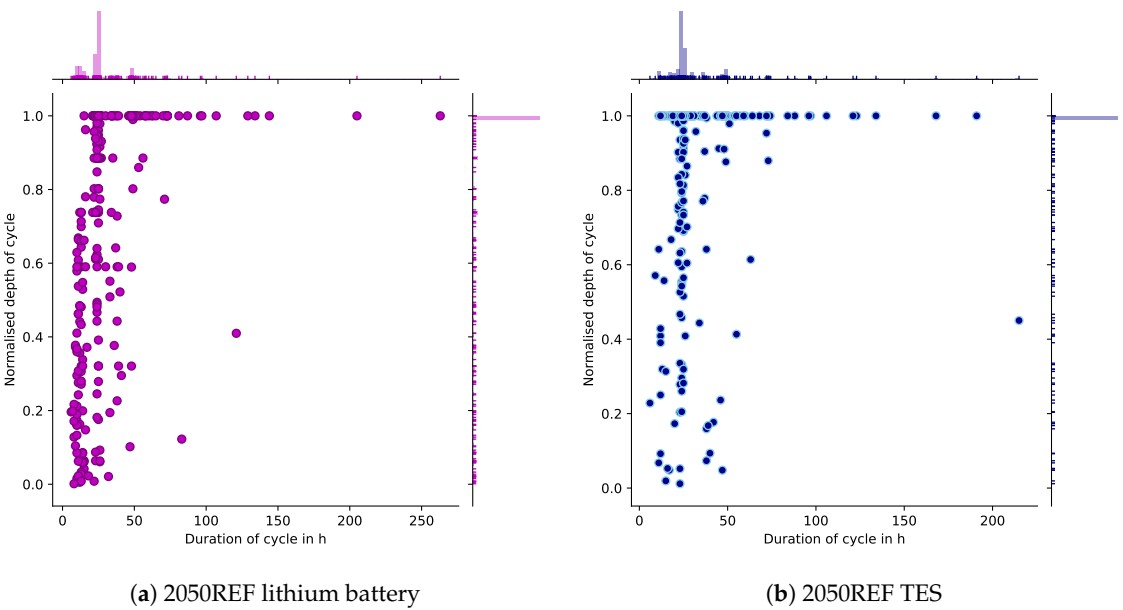

(**a**) 2050REF lithium battery

(**b**) 2050REF TES

**Figure 5.** Storage cycles of lithium battery and TES for the 2050 scenario.

Figure 6 shows the temporal operation of TES during the year (a) and caused differences in storage charge and discharge for lithium storage units (b) due to heat pump operation with and without TES. The temporal impact on electricity storage operation can be analysed in Figure 6b. The plot reveals a clear seasonal and daily pattern. Charging is reduced during the summer months at noon, when PV generation peaks. In contrast, discharge is reduced in evening times. Charging and discharging of the storage is reduced to over 9 GW in some hours of the year. This pattern shows the PV integrating effect of TES by replacing electricity storage units. Due to the cycles of the TES, impact on long-term storage operation is significantly lower compared to short-term storage operation, which is reflected in the investment as well.

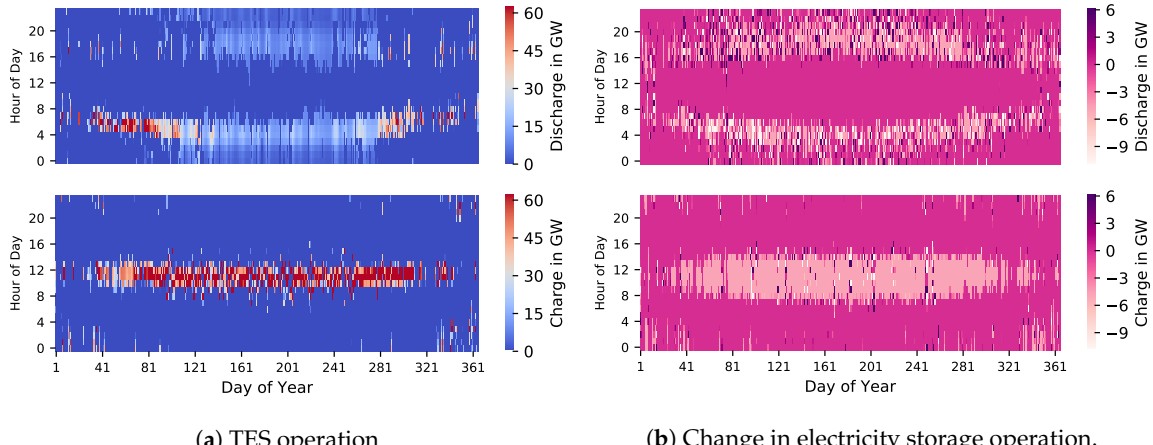

(**a**) TES operation

(**b**) Change in electricity storage operation.

**Figure 6.** Temporal operation of TES during the year (**a**) and caused differences in storage charge and discharge for lithium storage units (**b**) due to heat pump operation with and without TES.

From Figure 6a, it can be seen that the daily effect also applies for TES operation. For TES charging is mainly taking place around noon. In this case low heat demand allows to charge storage units with

PV during the day. In the summer discharge is lower and distributed for a longer period of time. Whereas in the colder month, discharge is shorter with a higher rate. In the main heating period, the pattern changes and charging is done at night instead during the day.

### 5.5. Sensitivities

The dispatchable biomass potential has a major impact on (electricity) storage investment [25]. Therefore, a sensitivity analysis has been conducted for the 2050 scenario with regard to the biomass potential. As shown in Figure 7, an increasing biomass share reduces battery as well as hydrogen storage investment.

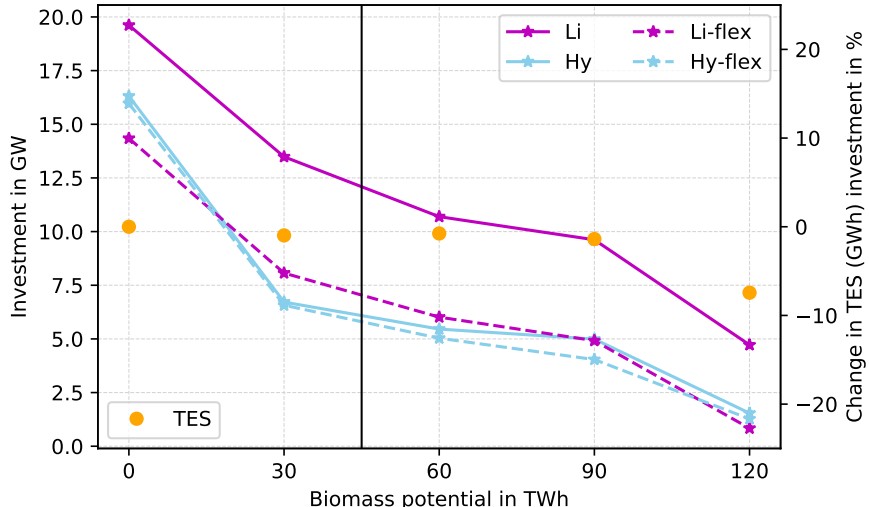

**Figure 7.** Sensitivity for installed storage capacities in GW in the 2050 scenario at different biomass potentials. Values for flexible heat pump operation are depicted as dashed lines. The *2050REF* scenario is indicated by the vertical line inside the figure.

At the same time, investment in TES is less sensitive with regard to the biomass potential. Without any biomass available, the hydrogen investment increases considerably by around 214% to 16 GW, while lithium increases to 19.6 GW by around 136% compared to the *2050REF* scenario. With a biomass potential of 120 TWh$_{th}$, investment decreases to 4.7 GW (lithium) and 1.5 GW (hydrogen). The effect of the HP flexibilisation is not effected substantially by the biomass potential. A reduction from 5.3 GW to 3 GW can be observed for the difference of flexible vs. in-flexible heat pump operation for short term storage units.

Figure 8 shows the results for the heat demand sensitivity. Clearly, electricity as well as TES investment increases with higher heat demand. While the TES investment changes linearly by about $\pm 20\%$, electricity storage units show a non-linear increase. Storage investment rises by 62.5% from 11 to 17.8 GW for lithium and by 67% from 5.5 to 9.1 GW for hydrogen with a 30% higher heat demand.

Nevertheless, according to the biomass potential, reduction in electricity storage investment due to heat pump flexibilisation is not affected substantially at different heat demands. Compared to the reference case (4.9 GW), short term electricity storage investment increases to 5.9 GW in the case of 30% higher demand. Similary, investment decreases to 4 GW in the case of 30% lower heat demand.

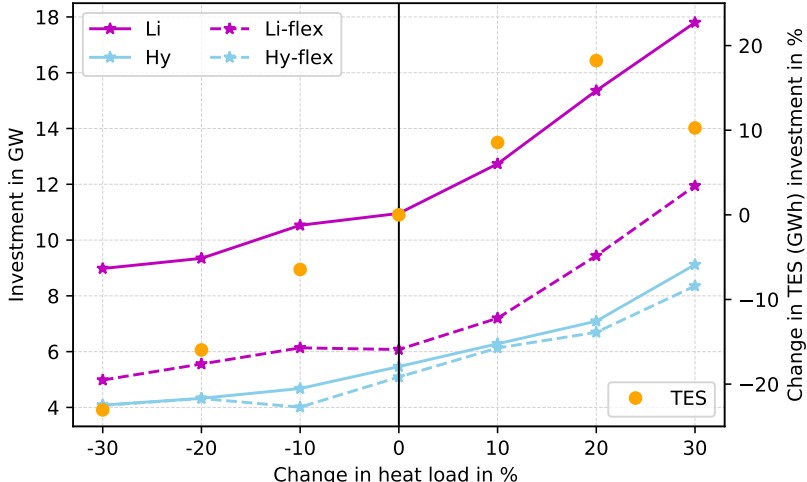

**Figure 8.** Sensitivity for installed storage capacities in GW in the 2050 scenario at different heat demand levels. Values for flexible heat pump operation are depicted as dashed lines. The *2050REF* scenario is indicated by the vertical line inside the figure.

## 6. Discussion

The presented results confirm the conclusions of Hedegaard and Münster [17], that the impact of heat pump flexibilisation is not relevant in systems with renewable energy shares below 80%. However, with higher RE shares, the importance of TES increases significantly. In addition, the conducted sensitivity analyses reveal the crucial role of available biomass potential for overall electricity storage investment. In particular long-term storage can by reduced more than half with the first 30 TWh of dispatchable biomass. Regardless, it has been shown that the flexibilisation of heat pumps is only slightly affected by the available biomass potential and therefore constitutes a robust option to reduce electricity storage requirements. Although the overall economic effect is small, indirect replacement of electricity storage investment by TES can be beneficial as less minerals like Cobalt, Lithium or Silver are required. Worldwide demand for Cobalt due to lithium batteries in 100% RE systems could exceed reserves even with high recycling rates and improvement in technologies. Similarly, lithium reserves may also be exceeded without high recycling rates [45], p. 446.

It is important to note that, due to the spatial resolution of the model, grid constraints inside countries are not modelled. Hence, storage investment might be required also in systems with less renewable penetration to ensure intact markets and avoid re-distpach. For example, the German grid development plan 2019 (German: "Netzentwicklungsplan") models scenarios of the electricity system with a RE share of around 67–68% in Germany and installed battery capacities of 8 to 12.5 GW [46] for 2030. Therefore, further investigations should include a higher spatial resolution including grid constraints of the transmission grid inside countries. With such a resolution, heat pump flexibilisation may become relevant even at lower renewable energy penetration.

Another aspect for discussion is the 100% RE scenario setting for the year 2050. This setting constitutes a scenario with an highly integrated European electricity system. In particular, Norway with large hydro capacities plays a crucial role in this scenario. While several studies have shown the benefits of integrated systems solutions opposed to island solutions, it is by no means clear that such scenarios actually materialise. Therefore, other 100% scenarios within less integrated systems should be developed to examine a broader spectrum of possible solutions. Nevertheless, the overall results indicate robustness for systems above 80% RE share.

As shown by Reference [25], DSM is an important option for renewable energy integration and can reduce electricity storage demand. Further research should cover interactions between heat pump flexibilisation, electricity storage and (electrical) DSM. As most electrical DSM options and the TES work on short time scales, the question of their combined potential arises.

## 7. Conclusions

This paper presents an open source model for Germany to analyse the interaction between investment in electricity storage and thermal energy storage units for heat pump flexibilisation in decentral heating systems. Overall, the results relate well to existing studies and show that TESs can help to integrate renewable energies by reducing electricity storage investment. In energy systems with a share of more than 80% renewable energy share the investment in short-term storage units can be reduced up to 42–62% by TES. Except for the 100% scenario, no investment in long term energy storages were observed. With a reduction of 0.37 GW (6.8%) the impact in this setting was comparatively low. Generally, storage investment increases significantly with reduced available biomass for dispatchable electricity generation. However, sensitivity analyses show, that the results of heat pump flexibilisation are rather insensitive with regard to the available biomass for electricity supply as well as to changes in the heat demand covered by heat pumps.

Overall, the results reveal only moderate need in additional short-term storage investment in the medium run in Germany. In particular, long term storage units like hydrogen are not required before renewable energy shares approach 100% of the electricity supply. With less than 1% reduction in system cost, the economic effect of flexible heat pump operation was found to be low. However, the indirect replacement of batteries with thermal energy storage units is environmentally beneficial due to a lower resource consumption of minerals. Therefore, heat pump flexibilisation can play an important role for a resource efficient energy transition.

**Funding:** This research was funded by Federal Ministry of Economic Affairs Germany grant number 03ET6122E.

**Conflicts of Interest:** The author declares no conflict of interest.

## Abbreviations

The following abbreviations are used in this manuscript:

| | |
|---|---|
| COP | Coefficient of Performance |
| FOM | Fixed Operation and Maintenance |
| HP | Heat Pump |
| IPCC | Intergovermental Panel on Climate Change |
| LP | Linear Programming |
| PHS | Pumped Hydro Storage |
| PV | Photovoltaic |
| RE | Renewable Energy |
| RoR | Run of River |
| TES | Thermal energy storage |
| TYNDP | Ten Year Network Development Plan |
| WACC | Weighted Average Cost of Capital |

## Appendix A. Model Symbols

**Table A1.** List of sets in the model.

| Symbol | Description |
|---|---|
| *C* | Set of all conversion processes |
| *D* | Set of all dispatchable generators |
| *H* | Set of all heat pumps |
| *K* | Set of all commodities |
| *L* | Set of all loads |
| *N* | Set of all transmission lines |
| *R* | Set of all reservoir units |
| *S* | Set of all storage units |
| *T* | Set of all timesteps |
| *V* | Set of all volatile generators |

**Table A2.** List of optimisation variables in the model.

| Symbol | Description |
|---|---|
| $x^{flow}(t)$ | Energy flow at timestep $t$ |
| $x_{h,to}^{flow}(t)$ | Heat flow to heat bus from heat pump $h$ timestep $t$ |
| $x_s^{level}(t)$ | Storage (energy) level of storage $s$ at timestep $t$ |
| $x_{h,to}^{capacity}$ | Thermal capacity of heat pump $h$ |
| $x_s^{capacity}$ | Capacity (power) of storage $s$ |
| $x_s^{storage\_capacity}$ | Storage capacity (energy) of storage $s$ |

**Table A3.** List of parameters in the model.

| Symbol | Description |
|---|---|
| $c_g^{mg}$ | Marginal cost of generator $g$ |
| $c_g^{cc}$ | Commodity cost of generator $g$ |
| $c_g^{vom}$ | Variable operational and maintenance cost of generator $g$ |
| $c_k^{amount}$ | Absolute amount of commodity $k$ |
| $c^{loss\_rate}(t)$ | Loss of storage energy per timestep $t$ |
| $c^{profile}(t)$ | Profile of generator, reservoir or load timestep $t$ |
| $c^{capacity}$ | Capacity of dispatchable or volatile generator $d$ / $v$ |
| $c_n^{loss}$ | Loss on transmission line $n$ |
| $c_e^{emission\_factor}$ | Emission factor of carrier $e$ |
| $c_c^{efficiency}(t)$ | Efficiency of conversion process $c$ at timestep $t$ |
| $c_s^{eta_in}$ | Charge efficiency of storage $s$ |
| $c_s^{eta_out}$ | Dis-charge efficiency of storage $s$ |

## Appendix B. Scenario Assumptions

*Appendix B.1. Residual Load*

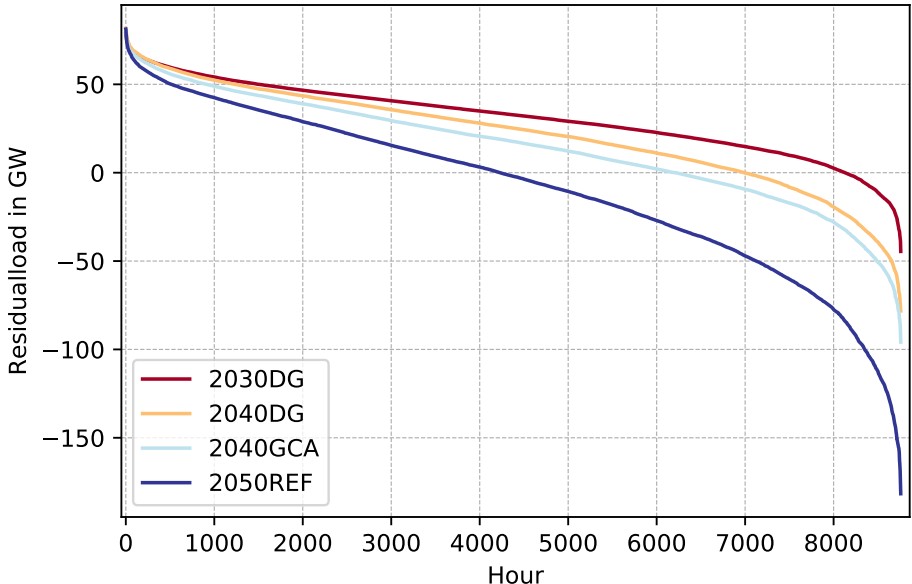

**Figure A1.** Electrical residual load in Germany within all main scenarios.

*Appendix B.2. Renewable Energy and Biomass Potentials*

**Table A4.** Full load hours of onshore, offshore, pv and run of river (RoR) supply.

| Country | Offshore | Onshore | PV | RoR |
|---|---|---|---|---|
| AT | - | 1507 | 1291 | 3058 |
| BE | 3939 | 2406 | 1135 | 1335 |
| CH | - | 1354 | 1416 | 3832 |
| CZ | - | 1875 | 1226 | 1974 |
| DE | 3976 | 1951 | 1151 | 4043 |
| DK | 4224 | 2670 | 977 | - |
| FR | 3295 | 2040 | 1265 | 2722 |
| LU | - | 2917 | 1192 | 2644 |
| NL | 4025 | 1921 | 1095 | 1518 |
| NO | 4341 | 3562 | 811 | 2028 |
| PL | 3964 | 1834 | 1113 | 1493 |
| SE | 3792 | 2654 | 862 | 2161 |

**Table A5.** Biomass potential of agriculture and forest residue per country in 2050 based on the *hotmaps* project [41]. For consistency German potential for electricity has been adapted with regard to the RESCUE study assumptions.

| | AT | BE | CH | CZ | DE | DK | FR | LU | NL | NO | PL | SE |
|---|---|---|---|---|---|---|---|---|---|---|---|---|
| Amount in TWh | 23.61 | 8.08 | 0.0 | 32.78 | 45.05 | 13.56 | 149.56 | 0.61 | 2.81 | 0.0 | 71.36 | 86.75 |

*Appendix B.3. Grid Capacities*

Figure A2 shows the installed the transmission capacities of the electricity system and pumped hydro storage capacities for all scenarios. As described above, the transmission system is modelled with a transshipment approach. The e-Highway 2050 in Figure A2d scenario includes major grid expansion to Scandinavian countries and the south east while the other scenarios only differ within a narrow range.

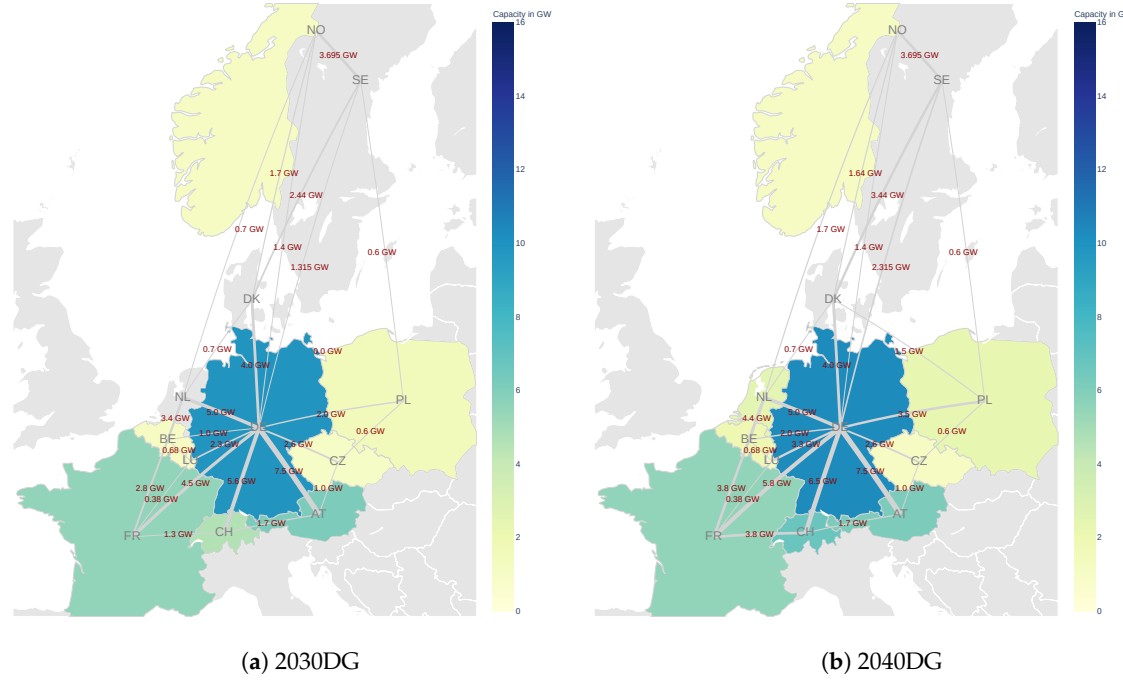

(**a**) 2030DG　　　　　　　　　　　　　　　　　　　(**b**) 2040DG

**Figure A2.** *Cont.*

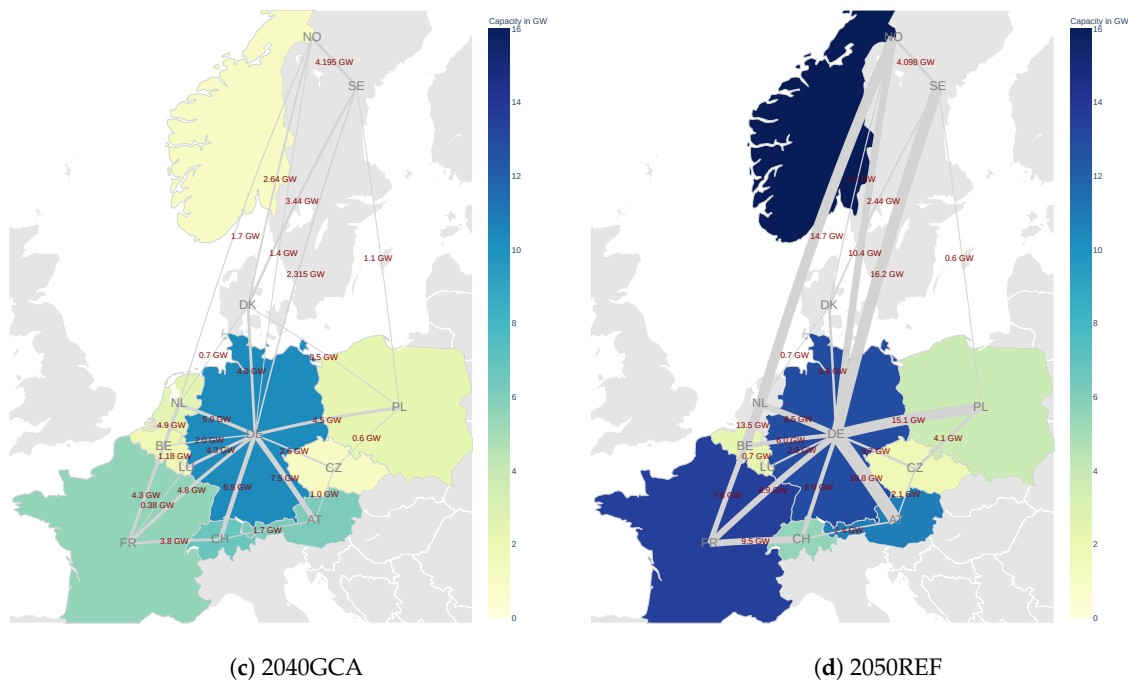

(**c**) 2040GCA                                      (**d**) 2050REF

**Figure A2.** Transmission and PHS storage capacities. Countries are dyed based on their installed PHS capacity in each scenario. The 2050 scenario is based on the *e-Highway2050* [29] 100% RES scenario. All other scenarios are based on the TYNDP2018 [31].

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
