# Peer review of "Effects of Decentral Heat Pump Operation on Electricity Storage Requirements in Germany"

_energies, doi:10.3390/en13112878_

Round 1
Reviewer 1 Report
The paper is interesting and well-written. Before publication it needs a few improvements.
1) Fig. 2 should be improved - solar and hydro- are not seen now.
2) Small English correction is advaiced for example - line 286: time instead of the time
3) Fig.6a is not well seen.
4) I recommend to extend paper by any Conclusions part.
Author Response
Dear Reviewer,
please find the updated manuscript attached and my answers to your comments below.
1) Fig. 2 should be improved - solar and hydro- are not seen now.
Thanks, the colour has been changed, also hydro capacities have been summed to be better visible. The problem here is, that hydro reservoir capacities in Germany are below 1GW, which can not be seen in the graphic well. Summed with run of river, capacities are around 5 GW, which is now hopefully better visible.
2) Small English correction is advised for example - line 286: time instead of the time
Thanks for the advice.
3) Fig.6a is not well seen.
Colours of the TES figure have been adapted to be seen better.
4) I recommend to extend paper by any Conclusions part.
A conclusion part has been added.
Kind regards,
Simon Hilpert

Reviewer 2 Report
This very interesting paper investigates the effect of decentral heat pump flexibilisation through thermal energy storage units on electricity storage investment. The analysis is carried using an open source model for the German electricity system based on the Open Energy Modelling Framework.
The introduction is too general to start off and lacks in data. It needs to set the scene more clearly. It needs to go straight to the topic and stay focused on it. In the 2nd paragraph of the introduction, the argument presented for the need for heat pumps is not convincing. It lacks data and a solid case.
The literature review is clear and concise. Relevant studies have been examined. Weaknesses have been identified.
The methods section is well thought out. Great to see the details of the modelling presented in full. Well thought out.
The scenarios and results have been presented clearly and show some interesting findings.
The significance of the findings has been discussed in a critical light and address the research objectives.
Overall, a good paper. Nice work.
Author Response
Dear Reviewer,
Thanks for your comments and your overall positive feedback. Please find the updated manuscript attached and my answers to your comments below.
Yes, its right the introduction start out with a wider scope. However I think it gets straight to the point by stating: 1) due to climate change the heating sector needs to be decarbonised 2) this is difficult to limited resources for heat 3) heat pumps therefore will have to play a major role 4) if so, the paper asks "what is the effect of flexible vs. no flexible operation. To me this gets straight to the point. I would therefore like to leave the introduction like it is. Also as other reviewers did not refer to the introduction.
However, literature has been added to support the importance of heat pumps in Germany in future energy systems. However, it should be noted, that the study investigates scenarios (if...then) to analyse what effects exists if there are many heat pumps. It does not say anything about the likely hood of the heat pump development. Nevertheless, I think the literature presented in the paper shows that the current state of scientific discussion indicates a great importance of heat pumps in renewable energy systems in Germany and Europe.
Kind regards,
Simon Hilpert

Reviewer 3 Report
Review of the MS “Effects of Decentral Heat Pump Operation on Electricity Storage Requirements in Germany” by Hilpert S.
In the present work the authors studied the effect of decentral heat pump flexibilisation through thermal energy storage units on electricity storage investment. The study was carried using an open source model for the German electricity system based on the Open Energy Modelling Framework. The author showed the importance of flexible heat pump operation in 100 % renewable energy system. Also, flexibilisation of heat pumps in the German energy system can significantly reduce the need for electricity storage units.
Discussion and interpretation of the results is good. The MS organization is average, should be little bit improved (see below). The practical merits of the MS are very good. The description is good and easy to understand. The MS is an good contribution to the field. The MS is therefore worth publication.
I have only a few comments and suggestions:
Comments:
- First two sentences should be deleted or moved to Introduction section.
- I think the author should include Conclusions as a separate section. This is extremely important for readers. Unfortunately, most readers are starting reading the Abstract thenflip through the article to look at the figures and the Conclusions. Conclusion should be written very clearly.
Recommendation: The MS can be published after minor revision.
Author Response
Dear Reviewer,
thanks for the feedback. I will include a conclusion section as advised.
Are you referring to the first two sentences of the discussion? I agree that the first sentence can be deleted. However, I think it might be worth stating at the beginning of the discussion that results relate well to existing studies. I would therefore remove the first sentence and keep the second one.
Kind regards,
Simon Hilpert
Reviewer 4 Report
Il paper require the modigfies reported in yellow in the attached file
Insert the symbology used in the formulas reported in the paper.

Author Response
Dear Reviewer,
thanks for you feedback and comments. Please find the updated manuscript attached and my answers to your comments below.
Some valuable comments have been used to adapt the manuscript accordingly:
- The table captions have been moved above the table.
- The sentence in in L301 for Figure 6 has been included.
- "(Equation 2)" has been added in L120
- Tables with list of symbols have been added to the Appendix.
However with regard to other comments, the manuscript has been left unchanged:
- The sentence added for Figure 3 can already be found in the beginning of section 5.1 (first sentence), L231.
- Likewise, the first sentence of section 5.4 gives the information about the content of the Figure . However based on the comment, this sentence has been adapted slightly.
- Additional suggested literature does not to support the content inside the paper where it has been added. Therefore it has not been included. The first paper is about an analyses of the performance of a DSHP experimental prototype, installed as the air-conditioning system. However, literature at that argument is about 100\% renewable energy systems scenarios up to 2050 where results indicate an important role of heat pumps. The other two paper added deals with an analyses of refrigerators, which is not subject of this study. Nevertheless two more studies have been added in the introduction L40), to support the argument of heat pump importance in Germany.
Kind regards,
Simon Hilpert
